# Biomechanical Analysis in Five Bar Linkage Prototype Machine of Gait Training and Rehabilitation by IMU Sensor and Electromyography

**DOI:** 10.3390/s21051726

**Published:** 2021-03-02

**Authors:** Jeong-Woo Seo, Hyeong-Sic Kim

**Affiliations:** 1Future Medicine Division, Korea Institute of Oriental Medicine, Daejeon 34504, Korea; jwseo02@kiom.re.kr; 2HUCA System Inc., Daegu 41061, Korea

**Keywords:** machine of gait training and rehabilitation, five-bar linkage, gait analysis, IMU sensor, electromyography

## Abstract

The prototype machine of gait training and rehabilitation (MGTR) with a five-bar linkage structure was designed to improve the common end-effector type. Additionally, the study was conducted to evaluate the joint angle and muscle activity during walking for the evaluation of prototype: (1) Background: The gait rehabilitation systems are largely divided into exoskeletal type and end-effector type. The end-effector type can be improved a gait trajectory similar to normal gait according to this prototype. Therefore, a new design of prototype MGTR is proposed in this study. (2) Methods: The gait experience was conducted with thirteen healthy male subjects using an inertial measurement unit (IMU) sensor and electromyography (EMG). It was compared that the hip and knee joints and the muscle activity between the normal gait and MGTR. (3) Results: The results showed that there was a high correlation between the knee joint angle for normal gait and MGTR. The range of motion (RoM) was small for the MGTR. The EMG results showed that the activation of the rectus femoris muscle was most similar to the normal gait and MGTR. (4) Conclusions: The characteristics of the kinematic variables of the subjects varied widely. It is necessary to modify the machine so that the link length can be adjusted in consideration of various segment lengths of patients.

## 1. Introduction

The gait is a basic mobility function. Restoring motor function in stroke patients involves the restoration of mobility and gait. Gait rehabilitation is applied based on the severity of the patient’s injuries. If the severity is high, the therapist will apply treatment to move the joint using minimal muscle strength through continuous passive motion (CPM) rehabilitation. Then, body weight-supported treadmill training (BWSTT) is applied for rehabilitation when the recovery is above a certain level [1]. BWSTT involves walking on a treadmill in an unloaded condition using a harness and has the advantage of intensively training with exercises that are similar to walking conditions [2]. The gait rehabilitation is based on the neural plasticity theory of activation of the cerebral cortex and reconstruction of motor neurons in place of the motor function of the damaged or missing brain [3]. The CPM and BWSTT treatment performed according to the neural plasticity theory are important for a physical therapist’s physical effort and proficiency [4]. 

The MGTR (machine of gait training and rehabilitation) was developed to overcome this limitation. These are divided into various types and applied clinically. It is divided into an exoskeletal type and end-effect type based on their operating style. A representative exoskeletal-type is Lokomat from Hokoma Inc. (Volketswil, Switzerland), which is a combination of a harness and treadmill, and walking is performed by attaching a skeletal machine to a lower limb. The trajectory and angle of the joint are similar to that of the normal gait. The purpose is to control the posture according to the position of the joint and guidance torque in order to achieve a similar gait pattern based on brain plasticity. In addition, representative end-effector types are the G-eo system from REHA technology (Olten, Switzerland) and the Gait trainer GT from REHA STIM (Berlin, Germany). The MGTR fixes the footplate, which is the end of the body, in the unloaded state. The movement of the ankle joint is similar to that of the ankle joint of subjects. The link structure also generates movement of the knee and hip joint. Hidler et al. [5] suggested that the kinematic trajectory of Lokomat is similar to that of the treadmill when compared to the normal gait. The results of previous studies suggest that the end-effector type is more effective for gait rehabilitation than that of the exoskeletal type despite the end-effector’s failure to implement a normal pattern. In Mehrholz and Pohl’s [6] systematic review was performed on the Gait trainer and Lokomat training effects for stroke patients. Both types could influence the outcome of gait rehabilitation after a stroke. The end-effector type showed a higher effect than that of the exoskeletal type. There could be a reason for these results in the guidance of movement of both devices. The exoskeletal type controls the knee and hip joints of the lower extremities. However, only the ankle joint is controlled for the end-effector type to implement the gait pattern. The patient is required to consciously prevent hyper-extension of the knee joint and maintain a walking trajectory [7]. The MGTR should be developed that combines the unintended prevention of hyperextension of the exoskeletal type and the intended prevention of the end-effector type. 

In this study, a new prototype machine of gait training and rehabilitation is proposed. In Section 2, a five-bar link structure mechanical design method is proposed. The common end-effector type is designed to drive the ankle joint control to the footplate connected from the rear single-bar linkage. It has limitations in implementing the foot trajectory similar to normal gait. This structure ensures that the lower limbs are not constrained to the machine. Additionally, it is possible to implement a pattern similar to normal walking. This prototype did not apply the actuator driven active assistance, it operated passively. In Section 3, the hip and knee joints and the muscle activity between normal gait and MGTR were compared to confirm the kinematic characteristics of the MGTR. The characteristics of rehabilitation using the five-bar link type in the prototype work were confirmed and to find what needs to be improved.

## 2. Methods

### 2.1. Development of MGTR 

#### 2.1.1. Mechanical Design

This machine of gait training and rehabilitation consists of a left and right side, stride control section, five section link, and footstep section. A flywheel is used for continuous drive. When the stance phase from heel-strike to toe-off of walking is performed, a swing on the opposite side occurs simultaneously. The five-bar link consisting of a crank, rocker, and coupler 1 and 2 is designed to construct the gait trajectory of the foot by adjusting the link length of the link connected to the end of coupler 2 by rotating the crank (Figure 1). The footplate manually touches the treadmill and bottom to allow the belt to rotate.

#### 2.1.2. Implementation of Gait Trajectory

In order to implement the trajectory of the foot during walking, the movement design of the five-bar linkage mechanism is mathematically written as Equation (1). The length ratio and drive joint spacing are considered when designing the five-bar link. To reduce the mechanical singularity, the RoM (range of motion) should be determined be setting the applicable link ratio and drive joint spacing. The RoM is a pattern of gait trajectories from the foot [8]. In this study, the position trajectory of the ankle was measured during walking with a 3D motion analysis system composed of infrared cameras and was mathematically modeled using a Fourier function (Equation (1) and Table 1). The mechanism was designed using the cost function to minimize the error of the trajectory (Gi) of the modeled ankle and the trajectory (Ti) of the five-bar linkage mechanism. The angle between coupler 1 and 2, the length of each link, and the position P(x, y) of the rocker pivot were set as design variables (Equation (1)). According to Equations (1) and (2), the foot trajectory during walking is presented in coupler 2, which is caused by the pivot of the rocker and crank (Figure 2).
(1)$$f\left(k\right)=a_{0}+\overset{5}{\underset{i=1}{\sum }}\left\{a_{i}\cos\left(i\omega t\right)+b_{i}\sin\left(i\omega t\right)\right\}$$
(2)$$f_{cost\left(\theta_{coupler},L_{crank},L_{coupler1},L_{coupler2},L_{rocker},P\right)}=\overset{n}{\underset{i=1}{\sum }}\|T_{i}-G_{i}\|$$

### 2.2. Biomechanical Analysis

#### 2.2.1. Subjects

The subjects were thirteen young males without musculoskeletal and neurological disease and medical history (Age: 25.54 ± 3.25 years, Height: 177.11 ± 5.88 cm, Weight: 81.67 ± 9.45 kg). We conducted a survey about the subjects’ medical history, and consent was received before the experiment. This study was conducted after the approval of the Institutional Review Board of the Korea National Rehabilitation Center (No.NRC-2017-04-029).

#### 2.2.2. Experimental Protocol

The scene of the experiment is shown in Figure 3. Two IMU sensors were attached to the front of the thigh and shank of the right leg to calculate the joint angle [9]. Four sensors for EMG were attached to the front and back thighs (Rectus femoris (RF) and Biceps femoris (BF)) and the shin (Tibialis anterior (TA) and Gastrocnemius medial (GM)).

The experimental method was explained to the patient before the experiment. First, the participants walked back and forth a distance of 10 m for 2 min. After resting, the subjects passively walked without assistance of motor driving at the MGTR for 2 min using the same method. The preferred walking speed was maintained during walking. 

#### 2.2.3. Measuring Equipment

The IMU with EMG all-in-one system (Delsys Trigno Avanti EMG with IMU sensor wireless system, Natick, MA, USA) was used to measure the kinematic data and EMG simultaneously. This wireless and flexible system is a measurement of a high fidelity surface EMG signal (EMG sensor specification: no inter-sensor latency (<1 sample period), bandwidth 20–450 Hz, a maximum sampling rate of 2000 sample/s, baseline noise of 750 nV RMS, CMRR > 80 dB, 16-bit EMG signal resolution). Additionally, provides motion detection signals through an onboard IMU composed of 3D Accelerometer, 3D Gyroscope, and 3D Magnetometer, integrated 3-DOF (built-in 9-DOF). The measuring settings were as follows. The sampling frequency was 150 Hz at IMU and 1112 Hz at EMG [10]. The resolution depth across the input range of EMG and IMU was 16 bits. The bandwidths were 50 Hz at IMU and 20–450 at EMG. The EMG sensor Butterworth filter bandwidth was 2 pole high pass corner, 4 pole low pass corner in Hz.

#### 2.2.4. Data Processing

The data used in the analysis were an average of a min of data except for 30 s at the beginning and end. The kinematic data from the IMU were passed through a second-order zero lag Butterworth filter with a cutoff frequency of 10 Hz. The EMG data were passed through a fourth-order zero lag Butterworth filter using a 25–200 Hz band-pass in order to eliminate noise [11]. 

The gait event was defined as 0 to 100% from the heel contact to the next heel contact. The heel contact was identified as the negative peaks of medial-lateral angular velocity obtained from the gyroscopes occurring in heel contact before the instant of minimum peak of anterior-posterior acceleration [12]. 

The joint angle calculated from accelerations and angular velocity. Two sensors were using the same plane, and zero reference coordinator. The acceleration was used for finding flexion angles, and gyroscope to eliminate the effect of vibrations on the accelerometer [12]. The hip joint angle is the angle between the gravity vector and a perpendicular vector to the femur. This vector is equal to the sense of gravity by the accelerometer. The knee joint angle is the absolute shank angle measured by the sensor of the shank and it is the angle between the gravity vector and a perpendicular vector to the tibia [13]. 

The sensors for joint angle obtained by IMU and EMG sensor were obtained by the same system. Resampling between IMU and EMG data were performed using spline in accordance with the same starting point in order to match the sampling frequency.

The maximum voluntary contraction (MVC) was measured to obtain accurate muscle activity [14]. The variables selected from the kinematic data were the RoM, maximum and minimum values for one cycle of gait event, and percent of MVC (%MVC), and the maximum, minimum, and integral EMG (iEMG) values were extracted from the EMG data. 

The MATLAB R2019a (Mathworks Inc., Natick, MA, USA) and SPSS ver. 24 software (IBM Inc., Armonk, NY, USA) were used for the data analysis and statistical analysis with a Wilcoxon signed rank test and Spearman’s rho correlation. The significance level was α = 0.05.

## 3. Results

### 3.1. Results of Kinematics

The hip and knee angle of the subjects were obtained from the data measured during normal walking and the gait event when boarding the MGTR. A correlation analysis was performed to confirm the statistical similarity of the changes of the knee and hip joint angle. The results showed a high correlation (Spearman’s rho = 0.86 **) for the knee joint angle and a low correlation (Spearman’s rho = 0.32 **) for the hip joint angle (** *p* < 0.01).

The maximum and minimum angles and the RoM of each subject were extracted. It was compared statistically to determine the motion differences between the normal gait and MGTR. There was a significant difference in the minimum angle, maximum angle, and joint motion range (RoM) in the hip joint (Table 2).

### 3.2. Results of Electromyography

The mean values of the muscle activity of the subjects in each interval of the data measured at the time of normal walking and the gait event with the MGTR walking were measured. Correlations between the mean values were analyzed to confirm the similarity of the changes in the muscle activity. There was no correlation between TA (Spearman’s rho = 0.00) and BF (Spearman’s rho = −0.17), a low correlation with GM (Spearman’s rho = 0.23 *), and a high correlation with RF (Spearman’s rho = 0.7 **) (*: *p* < 0.05, **: *p* < 0.01, α = 0.05).

The statistical comparisons of the Wilcoxon signed rank test were conducted to determine the muscle activity characteristics of the subjects measured during general walking and MGTR boarding. There was no statistical difference in all cases of the value of the maximum and minimum, location of the maximum and minimum, and integrated muscle activity (iEMG). The maximum and minimum values and the iEMG were similar; however, the location (or timing) of the maximum and minimum values differed in all cases (Table 3).

## 4. Discussion

In this study, the new prototype machine of gait training and rehabilitation was developed, and gait analysis was performed to confirm the difference from normal walking by IMU and EMG sensor. The kinematic characteristics of the passenger during the five- bar linkage type passive MGTR were verified by mathematical modeling. A typical end-effector type consists of a single-bar linkage connected to the rear of a footplate, which implements a walking trajectory and a pattern with a motion with one degree of freedom. The prototype was designed as a five-bar linkage with two degrees of freedom, because it cannot constrain the movement of the link to only one input from the five-bar linkage structure. Generally, when the sagittal plane is viewed, the motion of the elliptical foot trajectory is generated. The distance of the ellipse is defined as the stance and the swing phase, and the height of the ellipse is a dorsi- and plantar-flexion around the ankle joint. This can be implemented with a motion similar to that of the five-bar linkage [15]. 

A kinematic motion analysis of this system and normal gait was performed (Figure 4). The knee joint angle showed a high correlation between the MGTR and normal gait. Therefore, the similarity of the patterns was high. In the swing phase, knee flexion showed a similar size and pattern to the normal gait. The gait in the MGTR was performed in the flexion state, and a knee extension did not occur in the stance phase. The knee maintained the flexion state from the heel strike to the initial contact of the heel. The hip joint was also confirmed to be in the flexion state, and hip joint extension did not occur. This could be verified by comparing the RoM results. The hip joint RoM of the MGTR was approximately 1/3 that of the normal gait, and the knee joint RoM was 1/2 that of the normal gait. This was similar to the results of previous studies that performed a kinematic analysis of an end-effector type. Previous studies have confirmed that the RoM of each joint is reduced by percentage of step length, because the lower limb is not completely restrained [16]. The main reason for this is that the prototype MGTR operates passively. The joints can be exerted more effectively when they are in a small flexion state rather than in a full extension state [17].

The RoM of the joint and its shape can be explained by the activation patterns of the muscles. The activation patterns of the four leg muscles were compared. There was no similarity at TA and BF for the MGTR and normal gait patterns. There was a correlation between GM and RF. 

The TA is mainly activated when dorsi-flexion of the ankle occurs. In the gait event, the initial stance and the mid-swing phase correspond to this [18]. The maximum activity was approximately 30% of the MVC, and the pre-swing and initial-swing events were approximately 61% of the gait events. The MGTR has a maximum activity of approximately 31% compared to the MVC and the mid-stance section, where the left and right feet correspond to 37% of the gait event intersect, based on the sagittal plane. The gait event involves the pull of the footplate from the back to the front. The minimum activities were similar at 6% and 7%, and the timings were 46% and 47% of the normal gait and MGTR, respectively.

The GM is activated in the stance phase during normal walking [19]. The maximum activities were 28% and 21%, and the timings were 33% and 38% of the normal gait and MGTR, respectively. The minimum activities were 3% and 2%, and the timings were 67% and 63% of the normal gait and MGTR, respectively. These were activated at a similar timing.

The RF was mainly activated in the terminal stance around knee extension, and there was no significant difference in the maximum activity and timing of the normal gait and MGTR (Figure 5). The maximum activities were 41% and 48%, and the timings were 50% and 39% of the normal gait and MGTR, respectively. This correlation was higher than that of the other muscles in the correlation showing pattern similarity. Although knee extension did not occur in the MGTR, the amount and pattern of the muscle activity were similar to those of the general gait. The RF is mainly activated from the terminal swing to the initial contact, as shown in the results of MGTR of the mean value based on each event timing in Figure 4. In addition, the RF is activated at the average maximum value and timing of the individual subjects, as listed in Table 3. It is similarly activated at the double support task phase of the gait, which is switched from the stance to the swing phase. The RF is activated at both sections of the normal gait, and the acceleration required to move the upper body occurs at the initial contact from the terminal swing [20]. The initial acceleration or propulsion was similar to the MGTR and normal gait, which is an important factor in gait rehabilitation [21].

The BF is activated before and after the heel contact event of the gait. In this study, the activation level was similar; however, there was a difference at the maximum timing. The maximum activities were 31% and 29%, and the timings were 61% and 41% of the normal gait and MGTR, respectively.

The results of iEMG were similar between the normal gait and MGTR. The MGTR activity was higher only in the RF muscle.

The kinematic results of the MGTR and normal gait have a larger standard deviation. The reason is that despite the large difference in the height of the subjects, the stride lengths of the MGTR were fixed. The trends of most subjects were similar for the normal gait, as shown in Figure 4. Accordingly, the standard deviation of the muscle activity was large for the individual subjects. Therefore, it is difficult to optimize the mean value of the MGTR boarding results and normal gait. The trend of the kinematic properties of the MGTR are more difficult to standardize than those of the normal gait because the lower limb is not constrained, as in the exoskeletal-type system, and the length of each link cannot be adjusted. The RoM is also different according to the difference of the thigh and shank length. As the prototype developed in this study maintained a fixed length of the link, a difference occurred. Therefore, the length around each leg joint and stride length should be adjusted to the subject in a revised version. In addition, the muscle activity should be examined when applying a power assist function because it is a passive-type system, in which torque is not applied.

Recently, an increasingly popular solution to improve gait abnormalities in hemiplegic patients is a wearable or exoskeletal type rehabilitation system [22,23]. The MGTR except exoskeletal type has limitations in considering the segment length of the patient. Therefore, there is a limit to patient specific personalized treatment. The MGTR mechanical design should be performed considering various factors such as analyzing the biomechanical characteristics of patients. Through further study will improve the problem of identified in this study such as adjustable length of the machine segment and apply joint assisted torque.

Additionally, the limitations of this study include the accuracy of the joint angle results. The IMU sensor drift was not considered when calculating the joint angle using the IMU sensor. There is a possibility that an error occurred because the joint angle could not be verified through comparison with the optical 3D motion analysis system. This issue had to be considered as an error identified in previous studies [13]. Furthermore, the synchronized issue was limited. It is necessary to minimize the error by optimizing the synchronization between the sensors through the same method as in previous studies since the data was measured by different sensors. It is necessary to review the accuracy and error of this system in subsequent studies [24].

## 5. Conclusions

We developed an adaptive end-effector-type MGTR prototype with a five-bar linkage drive system which is different from the existing end-effector type system. The differences in the hip and knee joint angles were confirmed by kinematic analysis, and based on the muscle activity. The activity of the knee extension muscle was similar to that of the RF muscle. The length of the segment was not being adjusted in this machine, and the length was difficult to quantify and optimize. It was confirmed that it is necessary to modify the mechanism capable of adjusting the link length in consideration of patients of various segment lengths through this gait movement analysis.

## Figures and Tables

**Figure 1 sensors-21-01726-f001:**
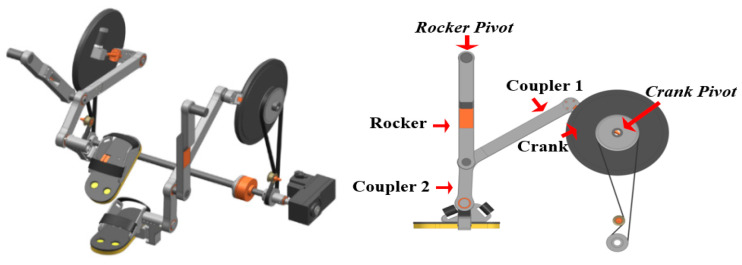
Concept of gait rehabilitation prototype and five-bar linkage mechanism.

**Figure 2 sensors-21-01726-f002:**
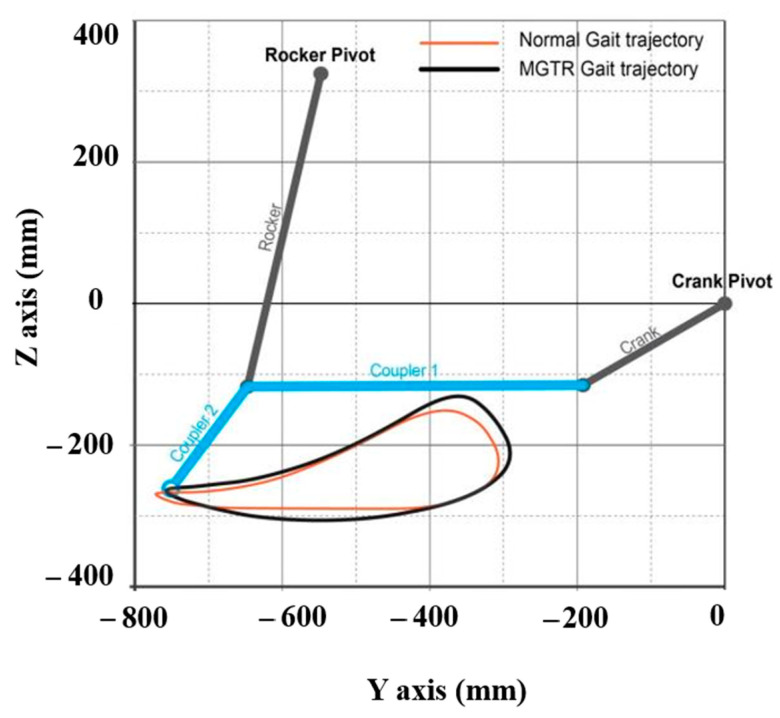
Result of implemented gait trajectory.

**Figure 3 sensors-21-01726-f003:**
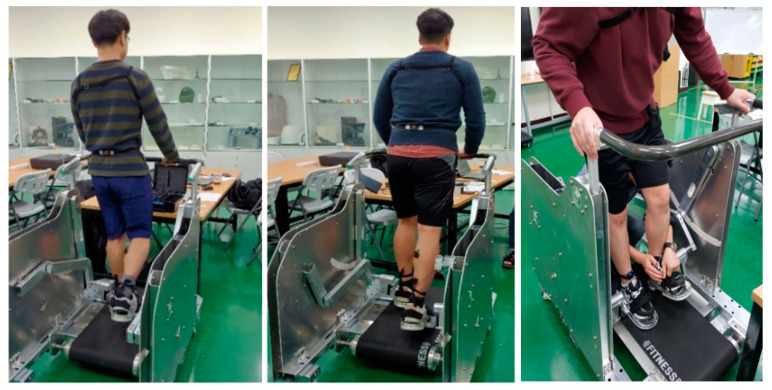
Scene of the experiment.

**Figure 4 sensors-21-01726-f004:**
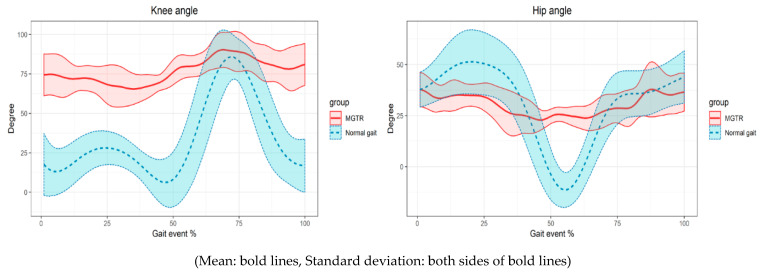
Results of the knee and hip angle with the gait cycle.

**Figure 5 sensors-21-01726-f005:**
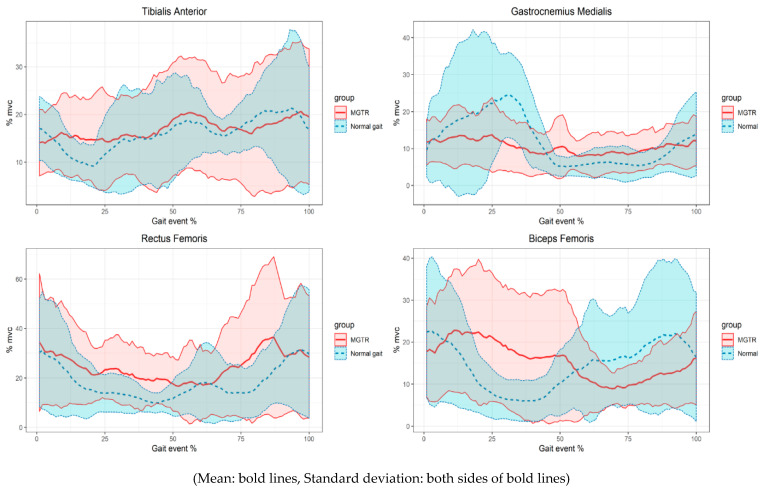
Results of the muscle activities with the gait cycle.

**Table 1 sensors-21-01726-t001:** Coefficients of Fourier fittings.

	$a_{0}$	$a_{1}$	$a_{2}$	$a_{3}$	$a_{4}$	$a_{5}$
Position x[m]	0.0316	0.2482	0.1426	0.0561	−0.0264	0.0092
Position y[m]	0.1341	0.0011	−0.0549	−0.0280	0.0033	0.0043
	$b_{0}$	$b_{1}$	$b_{2}$	$b_{3}$	$b_{4}$	$b_{5}$
Position x[m]	−0.0082	0.0054	−0.0040	0.0007	−0.0033	4.174
Position y[m]	25.54 ± 3.25	177.11 ± 5.88	81.67 ± 9.45	0.0088	−0.0002	4.174

**Table 2 sensors-21-01726-t002:** Results of the kinematic data.

		Min. Angle	Max. Angle	RoM
Hip	Gait	−14.56 ± 7.50	54.40 ± 12.59	68.97 ± 14.65
	MGTR	24.02 ± 3.16	43.32 ± 5.74	19.31 ± 3.11
	*p*	0.00 *	0.03 *	0.00 *
Knee	Gait	13.85 ± 6.66	93.81 ± 14.31	79.96 ± 19.25
	MGTR	60.17 ± 3.41	102.78 ± 6.90	42.61 ± 4.25
	*p*	0.00 *	N.S	0.04 *

*: Wilcoxon signed rank test (*p* < 0.05, *α* = 0.05), mean ± SD, N.S: Not significant.

**Table 3 sensors-21-01726-t003:** Results of the muscle activation variables.

Muscles		Max Value (%)	Event of Max (%)	Min Value (%)	Event of Min (%)	Iemg (∑)
TA	Gait	30.03 ± 11.89	61.05 ± 35.65	6.92 ± 2.67	46.06 ± 39.60	1600.18 ± 579.66
	MGTR	31.93 ± 6.84	37.50 ± 22.88	7.11 ± 4.01	47.75 ± 35.22	1647.64 ± 712.75
	*p*	N.S	N.S	N.S	N.S	N.S
GM	Gait	28.08 ± 10.38	33.40 ± 2.70	3.11 ± 0.77	67.20 ± 9.78	890.63 ± 210.31
	MGTR	21.18 ± 4.39	38.33 ± 51.68	2.51 ± 0.84	63.33 ± 36.50	898.39 ± 221.74
	*p*	N.S	N.S	N.S	N.S	N.S
RF	Gait	41.09 ± 26.75	50.92 ± 42.07	7.05 ± 4.38	59.33 ± 17.91	1798 ± 1053.62
	MGTR	48.25 ± 32.09	39.75 ± 42.08	8.26 ± 3.70	54.63 ± 29.35	2211.11 ± 1231.13
	*p*	N.S	N.S	N.S	N.S	N.S
BF	Gait	31.20 ± 18.96	61.83 ± 36.98	4.83 ± 4.04	40.83 ± 16.94	1436.70 ± 945.00
	MGTR	29.74 ± 19.16	41.20 ± 40.01	5.84 ± 4.19	54.90 ± 23.80	1420.32 ± 859.95
	*p*	N.S	N.S	N.S	N.S	N.S

*: Wilcoxon signed rank test (*p* < 0.05, *α* = 0.05), mean ± SD, N.S: Not significant.

## Data Availability

The data are not publicly available due to company security policy and personal protection of subjects.

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
