# Peer review of "Biomechanical Analysis in Five Bar Linkage Prototype Machine of Gait Training and Rehabilitation by IMU Sensor and Electromyography"

_sensors, 2021, doi:10.3390/s21051726_

Round 1

Reviewer 1 Report

The manuscript was submitted under "Communication", which confused me: why didn't you submit it to "Article"? What prevented you from doing so? It is because that you could not provide more details about the system and the experiments?

The contribution is very limited and questionable:

In the Abstract, the authors mentioned "The new prototype gait rehabilitation robot (machine of gait training and rehabilitation, MGTR) consists of a five-bar linkage system." I have questions regarding this first sentence:

  1. Why did you use "The", compared with what? 
  2. Please avoid creating some new name from commonly used terminologies in this field. It is adding unnecessary noise to the field and could not really differentiate your work from the others. In other words, please consider using another name (if you really want to create one) rather than "machine of gait training and rehabilitation")

As you said, "The MGTR prototype did not apply the active assistance, it operated passively." In my opinion, it is very inappropriate to call the system a "robot". The details of the system are very unclear - I could easily call it a stationary bicycle.

The IRB approval # seems to be four years ago (2017), could you explain why?

Author Response

Response to Reviewer 1 Comments

Point 1: The manuscript was submitted under "Communication", which confused me: why didn't you submit it to "Article"? What prevented you from doing so? It is because that you could not provide more details about the system and the experiments?

Response 1: Thank you for reviewing this manuscript. This manuscript was originally submitted as an “Article”. I received an answer from sensors journal saying that it is not an “Article” type paper. And It was recommended to submit to the “Communication”. This seems to be a matter of pages. (this manuscript is only nine pages).

Point 2: Why did you use "The", compared with what? 

Response 2: As can be seen from the results of this study, the kinematic data and EMG data that are found during normal walking were compared. It is not intended to verify the final completed system. It is a prototype system that can implement the gait ankle trajectory of the end-effect type with a five-bar linkage structure. The purpose is to identify kinematic variables that need to be improved for user boarding. Therefore, it was compared with normal gait of floor.

Point 3: Why Please avoid creating some new name from commonly used terminologies in this field. It is adding unnecessary noise to the field and could not really differentiate your work from the others. In other words, please consider using another name (if you really want to create one) rather than "machine of gait training and rehabilitation")

Response 3: As you pointed out, the “robot” word has been modified. And it was unified through “machine of gait training and rehabilitation” without creating new words.

Point 4: As you said, "The MGTR prototype did not apply the active assistance, it operated passively." In my opinion, it is very inappropriate to call the system a "robot". The details of the system are very unclear - I could easily call it a stationary bicycle.

Response 4: This is a study conducted in the basic stage. It will be finally developed as a robot. The “robot” word is modified by the system or machine.

Point 5: The IRB approval # seems to be four years ago (2017), could you explain why?

Response 5: The IRB of this study was acquired in April 2017 at the Korea National Rehabilitation Center. and the experiment of this study was conducted in January 2018. The manuscript was not written as a result of this study at that time. And then, there was a discussion of writing manuscript and submit with this research director recently. It was writing and submitted late for this reason. I am confident that there have been no problems such as experimentation and dissertation that violate IRB's research ethics.

Reviewer 2 Report

The paper contains a very valuable contribution. The subject is within the scope of the journal and the objective of research is well stated. However, some clarifications about the underlying hypothesis / scope are needed.

In the opinion of this Reviewer the manuscript deserves to be published once the Author takes into account the raised issues.

Abstract

  1. This reviewer suggests writing a more discursive abstract. It is too schematic

Introduction / Literature review

  1. The research scope is clear as well as the literature review. Anyway, the authors should better highlight the innovative aspects of their work in the manuscript, in particular in relation to passive structures.

What are the advantages / improvements in the proposed approach, which are not covered by the current studies?

  1. For the sake of readability, at the end of Section 1 the authors should describe how the paper is structured.

Methods - Experimental Condition and Protocol

  1. Please check the values of weight and height of the 13 young males subject used for the experiments. Probably the authors exchanged them.
  2. It is not clear to this reviewer the type of HW used. In particular, did the authors use the TRIGNO EMG wireless system only for the EMG? What did they used as IMU unit? Why did they choose 1112Hz for IMU and 145Hz for the EMG as sampling frequency? What is the full-scale range for the IMU. What is the accuracy of the entire system? Did the author use some technique for data fusion of the accelerometer, gyroscope, magnetometer (like complementary filter or similar)?
  3. How did the author synchronize the EMG with kinematic data? Synchronization is an important issue in such a system. Could the proposed solution gain better results with a multi-unit synchronized system for activity monitoring (e.g., https://doi.org/10.3390/electronics9071118, document that could be cited in the text)? How much does synchronization and measurement accuracy affect the entire system?

Results and discussion

  1. In the opinion of this reviewer, these two sections could be joined and expanded. It is very difficult to fully understand the discussion without some describing figures. This reviewer suggests adding more figures/graphs related to the single aspect/phase and before commenting it, adding a brief description of the phase itself.

Minor

  1. The authors should check that all the used acronyms are explained and not repeated every time (see CPM, MGTR, etc)
  2. Extensive editing of English language and style required. The paper should be carefully rechecked.

Author Response

Response to Reviewer 2 Comments

Point 1: This reviewer suggests writing a more discursive abstract. It is too schematic

Response 1: As you pointed out, the abstract was revised to be more discursive. The front part of the schematic form has been deleted, and the “Abstract” has been modified to clearly present the purpose of the study.

Point 2: The research scope is clear as well as the literature review. Anyway, the authors should better highlight the innovative aspects of their work in the manuscript, in particular in relation to passive structures.

Response 2: At the end of the introduction, the purpose of the study has been revised to be clearly presented.

Point 3: What are the advantages / improvements in the proposed approach, which are not covered by the current studies?

Response 3: I think that the improvement of this study is to understand the biomechanical gait characteristics of the users on board this new type system. The purpose of this study was newly implemented as a five bar link structure rather than a common single link structure end-effector type. The differences things were investigated and compared with normal walking by biomechanical method. This was supplemented by revising the end of the “introduction” part. (Line 65~75)

Point 4: For the sake of readability, at the end of Section 1 the authors should describe how the paper is structured.

Response 4:  At the end of Chapter 1, the structure of manuscript was explained. The introduction was modified that the “First. Device development”, then, “Second, Biomechanical analysis”. (Line 65~75)

Point 5: Please check the values of weight and height of the 13 young males subject used for the experiments. Probably the authors exchanged them.

Response 5: The weight and height of subjects were presented as mean and deviation values ​​in “Line 110”.

Point 6: It is not clear to this reviewer the type of HW used. In particular, did the authors use the TRIGNO EMG wireless system only for the EMG?

Response 6: The Delsys Trigno Avanti sensor used in this study is an integrated system with EMG and IMU. The hardware descriptions have been added to “Lines 114~123”.

Point 7: What did they used as IMU unit?

Response 7: The inertial measurement unit (IMU) sensor with electromyograms all-in-one system (Delsys Trigno Avanti EMG with IMU sensor wireless system, USA) were used to measure the kinematic data and EMG simultaneously.

Point 8: Why did they choose 1112Hz for IMU and 145Hz for the EMG as sampling frequency?

Response 8: The EMG and IMU systems are all-in-one, but the sampling frequency cannot be the same technically. This is systematical characteristic, and other previous studies also perform resampling after data acquired.

Point 9: What is the full-scale range for the IMU. What is the accuracy of the entire system?

Response 9: The Trigno Avanti™ Sensors have a built-in 9-DOF, one of 4 ranges can be selected for each sensor to span ±2g to ±16g for accelerometer outputs and ±250°/s to ±2000°/s for gyroscope outputs. You can check the level of accuracy through the EMG latency and “g” range of the IMU sensor. (Line 114-123)

Point 10: Did the author use some technique for data fusion of the accelerometer, gyroscope, magnetometer (like complementary filter or similar)?

Response 10: The gait event was defined as 0 to 100% from the heel contact to the next heel contact. The acceleration data were differentiated, and the event was calculated as the interval where the transition occurred. (Line 127-129)

Point 11: How did the author synchronize the EMG with kinematic data? Synchronization is an important issue in such a system.

Response 11: The EMG data were synchronized with the kinematic data, in which the gait event was confirmed, and data of the same interval were extracted. The method in “reference [10] was referred. (Line 128-130)

Point 12: Could the proposed solution gain better results with a multi-unit synchronized system for activity monitoring (e.g., https://doi.org/10.3390/electronics9071118, document that could be cited in the text)? How much does synchronization and measurement accuracy affect the entire system?

Response 12: The Delsys Avanti sensor is a device whose accuracy has already been proven through numerous studies. I believe that technically evaluating and verifying the accuracy of the sensor is a separate matter from the purpose of this study. The manuscript you suggested has been verified. I will conduct the same procedure at next similar study.

Point 13: In the opinion of this reviewer, these two sections could be joined and expanded. It is very difficult to fully understand the discussion without some describing figures. This reviewer suggests adding more figures/graphs related to the single aspect/phase and before commenting it, adding a brief description of the phase itself.

Response 13: The position of the figure in the “result” is placed in the “discussion” so that the readability can be improved. Adding a single aspect/phase is also a good idea. However, the results with individual differences are shown as representative graphs of mean and deviation. Therefore, we will consider whether presentation of personal data is absolutely necessary.

Point 14: The authors should check that all the used acronyms are explained and not repeated every time (see CPM, MGTR, etc)

Response 14: The unification and correction of words was applied. Thank you.

Point 15: Extensive editing of English language and style required. The paper should be carefully rechecked.

Response 15: English of this manuscript has been modified. I will improve the quality of English even better through extensive editing of English provided by the Sensors journal if the paper is confirmed to be published.

Round 2

Reviewer 2 Report

The authors have enriched their work. Many points may not have been clear and have not been addressed in a proper way.

  1. Previous point 4. I suggest describing the structure of the paper with the section numbers (eg. In the section 2 …., in the section 3…. Etc)
  2. Previous point 5. I think that the weight and height are inverted. 177 kg and 81cm? Probably you mean 177cm and 81kg.
  3. Previous point 8. The question was related to the number. I understand that the sampling frequency are different. The question is, why 1112Hz for IMU and why 145Hz for EMG? Usually for the IMU a sampling frequency of 100Hz is quite enough and some studies demonstrate that. Please justify those choices.
  4. Previous point 9. Also in this case, the question was not about the capability of the Trigno Avanti Sensors. The question is about the real settings used in the experiment, and why did you choose them.
  5. Previous point 10. It is not clear how did you use the data from accelerometer, gyroscope and magnetometer. What do you mean “The acceleration data were differentiated” at row 129? Do you use only the accelerometer for the gait identification? Or, do you use all the data for gait cycles calculation? Do you use only the gyroscope for the angle of the hip joint? What about the intrinsic drift of the gyroscope? Please give more detail about the sensor data manipulation.
  6. Previous point 11. The question is how the synchronization is made. I read the reference you cited, and the synchronization problem is note mentioned. Please add more details about it or add a suitable reference.
  7. Previous point 12. Probably for this point it was my fault. The main question was not about the accuracy of the measurement, but the accuracy of the synchronization system. The synchronization is a very big deal especially if you want to correlate measures taken with different and separate systems. I think that at least a mention to the problem and the possible solutions could be an added value to this manuscript.

Minor

  1. Check the parenthesis at rows 121-124

Author Response

<Response to Reviewer 2 Comments>

Point 1: Previous point 4. I suggest describing the structure of the paper with the section numbers (eg. In the section 2 …., in the section 3…. Etc)

Response 1: According to what you said, the “2. methods” and “3. results” are divided into sections (ex.2.1.1~2.2.4, 3.1~3.2).

Point 2: Previous point 5. I think that the weight and height are inverted. 177 kg and 81cm? Probably you mean 177cm and 81kg.

Response 2: It is my mistake not to check it. Corrected kg and cm. We apologize for not being able to check it thoroughly. Thank you.

Point 3: Previous point 8. The question was related to the number. I understand that the sampling frequency are different. The question is, why 1112Hz for IMU and why 145Hz for EMG? Usually for the IMU a sampling frequency of 100Hz is quite enough and some studies demonstrate that. Please justify those choices.

Response 3: As a result of checking the raw data again, it was confirmed that the sampling frequencies of the EMG and IMU were changed. EMG was collected at 1,112Hz and IMU at 150Hz. This is absolutely the authors' mistake. We regret making mistakes on very important things. We apologize for not being able to check it thoroughly. Below is a capture image of the raw data as evidence for the changes.(included in PDF file)

Point 4: Previous point 9. Also in this case, the question was not about the capability of the Trigno Avanti Sensors. The question is about the real settings used in the experiment, and why did you choose them.

Response 4: As mentioned in point 3, there was a mistake in the sampling frequency. And, does it mean the attachment position and selection to calculate the joint angle in the experiment? The information of experiment setting is reported that IMU were attached to the thigh and shank was simply written in lines 126-127. Also, added a reference [9] (IMU-based joint angle measurement for gait analysis. Sensors 2014) for choosing the method of attaching to two limbs to calculate the joint angle.

Point 5: Previous point 10. It is not clear how did you use the data from accelerometer, gyroscope and magnetometer. What do you mean “The acceleration data were differentiated” at row 129? Do you use only the accelerometer for the gait identification? Or, do you use all the data for gait cycles calculation? Do you use only the gyroscope for the angle of the hip joint? What about the intrinsic drift of the gyroscope? Please give more detail about the sensor data manipulation.

Response 5: I understood that there is a lack of explanation on how to calculate the gait event and joint angle using the IMU. Added explanation on how to define the gait event from gyroscope data at “Line 154-156”. ([11] Trojaniello, D. Estimation of step-by-step spatio-temporal parameters of normal and impaired gait using shank-mounted magneto-inertial sensors: Application to elderly, hemiparetic, parkinsonian and choreic gait. J Neuroeng Rehabil 2014, 11, 152.)

Also, added explanation on how to calculate hip and knee ankle angles from sensors attached to high and shank. This research method and appropriate preceding research are suggested as a reference for calculating the joint angle by IMU at “Line 157-163”. ([12] Bakhshi, S.; Mahoor, M.H.; Davidson, B.S. Development of a body joint angle measurement system using IMU sensors. Annu Int Conf IEEE Eng Med Biol Soc. 2011, 6923-6926). The issue of error due to sensor drift is a situation that needs to be considered in Bakhshi's paper. These limitations of these studies were mentioned at the end of the discussion at “Line 304-311”.

Point 6: Previous point 11. The question is how the synchronization is made. I read the reference you cited, and the synchronization problem is note mentioned. Please add more details about it or add a suitable reference.

Response 6: The sensors for joint angle obtained by IMU and EMG sensor were obtained by the same system. Resampling between IMU and EMG data were performed using spline in accordance with the same starting point in order to match the sampling frequency. (Line 164-166)

Point 7: Previous point 12. Probably for this point it was my fault. The main question was not about the accuracy of the measurement, but the accuracy of the synchronization system. The synchronization is a very big deal especially if you want to correlate measures taken with different and separate systems. I think that at least a mention to the problem and the possible solutions could be an added value to this manuscript.

Response 7: It seems to refer to the synchronization problem of EMG acquisition data and IMU data for joint angle measurement. Both data are measured by one system. Data measured by multiple channels in one system with the same start and end. It is not separated system. The sampling frequency is synchronized by resampling (spline method) to equalize the number of samples. Normal gait and MGTR gait are displayed in the range of 0 to 100 according to the calculated gait of walking events. As answered in Point 6, a paper as a reference [11] was presented.

Point 8: (Minor); Check the parenthesis at rows 121-124

Response 8: Parentheses have been corrected.

Round 3

Reviewer 2 Report

Thank you for the reply. However, some points still not have been addressed in a proper way.

  1. Previous point 3 about the sampling rate. Now that the frequencies used have been clarified, is it possible to have an explanation of these choices? Why 1112Hz and not 1000Hz for EMG? Is there a particular reason?

  1. Previous point 4. What I need to know is not only the sampling frequency. In such applications, the FSR, the bandwidth, the internal filtering, bit used for the representation are very important parameters. When you present the measuring equipment the sampling rate of 2000sample/sec (row 143) is reported. I understand that you are describing the equipment, but the same description is needed for the experimental setup.

  1. Previous point 6. The paper proposed in my previous review is an example of how to solve the problem of the synchronization with two or more different boards. In your experiment, you are using two different IMU sensors (row 126). My initial reference has always referred to this point. Please, comment it.

Author Response

<Response to Reviewer 2 Comments>

Point 1: Previous point 3 about the sampling rate. Now that the frequencies used have been clarified, is it possible to have an explanation of these choices? Why 1112Hz and not 1000Hz for EMG? Is there a particular reason?

Response 1: Thank you for your careful review. It can be seen that the frame interval is fixed at 13.5ms when looking at the Delsys Trigno Avanti sensor user's guide provided by Delsys Corp. Actually, it was impossible to set the sampling frequency to 1,000hz at the measurement. The 1,112Hz was the default when measured with Delsys “EMGworks Acquisition software”. I think this is a frame interval problem. There is no reason for measuring by 1,112Hz, it was just faithful to the system settings. A previous study [ref. No.10] that measured EMG at 1,112hz was presented as a reference in order to establish the evidence. Thank you.

Point 2: Previous point 4. What I need to know is not only the sampling frequency. In such applications, the FSR, the bandwidth, the internal filtering, bit used for the representation are very important parameters. When you present the measuring equipment the sampling rate of 2000sample/sec (row 143) is reported. I understand that you are describing the equipment, but the same description is needed for the experimental setup.

Response 2: It seems to be a misunderstanding caused by not using the word "maximum" sampling rate (modified). It has been described at the experimental setup. The measuring setting were as follows. “The sampling frequency were 150Hz at IMU and 1,112Hz at EMG. The resolution depth across input range of EMG and IMU were 16 bits. The bandwidths were 50Hz at IMU and 20-450 at EMG. The EMG sensor internal Butterworth filter bandwidth was 2 pole high pass corner, 4 pole low pass corner in Hz” (line 146-150) Thank you.

Point 3: Previous point 6. The paper proposed in my previous review is an example of how to solve the problem of the synchronization with two or more different boards. In your experiment, you are using two different IMU sensors (row 126). My initial reference has always referred to this point. Please, comment it.

Response 3: The synchronized method of this study was just resampling performed in post-processing. I know it was a limitation. Your previous study manuscript was commented at the end of the discussion as a reference although not confirmed in this study.

The comment was follows: “the synchronized issue was limited. It is necessary to minimize the error by optimizing the synchronization between the sensors through the same method as in previous studies since the data was measured by different sensors. It is necessary to review the accuracy and error of this system in subsequent studies [23]” (line 313-317)” Thank you.

Round 4

Reviewer 2 Report

Authors have properly enriched their work, by addressing each comment in a suitable way. The paper turns out to be notably improved.